# Paradoxical Pro-angiogenic Effect of Low-Dose Ellipticine Identified by In Silico Drug Repurposing

**DOI:** 10.3390/ijms22169067

**Published:** 2021-08-23

**Authors:** Jisu Oh, Hyeon Hae Lee, Yunhui Jeong, Siyeong Yoon, Hyun-Ju An, Minjung Baek, Do Kyung Kim, Soonchul Lee

**Affiliations:** 1Division of Hemato-Oncology, Department of Internal Medicine, Yongin Severance Hospital, Yonsei University College of Medicine, 363 Dongbaekjukjeon-daero, Giheung-gu, Yongin-si 16995, Korea; newfascia5@gmail.com; 2Department of Orthopaedic Surgery, CHA Bundang Medical Center, CHA University School of Medicine, Seong-nam 13496, Korea; aotcnlsl@gmail.com (H.H.L.); jeongyunhui92@gmail.com (Y.J.); tldud1105@naver.com (S.Y.); yks486ahj@naver.com (H.-J.A.); eclsa79@gmail.com (M.B.); 3CHA Graduate School of Medicine, 120 Hyeryong-ro, Pocheon 11160, Korea; fkzhfjqm@naver.com

**Keywords:** in silico, drug repurposing, noncytotoxic concentration, ellipticine, angiogenesis

## Abstract

Inadequate vessel maintenance or growth causes ischemia in diseases such as myocardial infarction, stroke, and neurodegenerative disorders. Therefore, developing an effective strategy to salvage ischemic tissues using a novel compound is urgent. Drug repurposing has become a widely used method that can make drug discovery more efficient and less expensive. Additionally, computational virtual screening tools make drug discovery faster and more accurate. This study found a novel drug candidate for pro-angiogenesis by in silico virtual screening. Using Gene Expression Omnibus (GEO) microarray datasets related to angiogenesis studies, differentially expressed genes were identified and characteristic direction signatures extracted from GEO2EnrichR were used as input data on L1000CDS^2^ to screen pro-angiogenic molecules. After a thorough review of the candidates, a list of compounds structurally similar to TWS-119 was generated using ChemMine Tools and its clustering toolbox. ChemMine Tools and ChemminR structural similarity search tools for small-molecule analysis and clustering were used for second screening. A molecular docking simulation was conducted using AutoDock v.4 to evaluate the physicochemical effect of secondary-screened chemicals. A cell viability or toxicity test was performed to determine the proper dose of the final candidate, ellipticine. As a result, we found ellipticine, which has pro-angiogenic effects, using virtual computational methods. The noncytotoxic concentration of ellipticine was 156.25 nM. The phosphorylation of glycogen synthase kinase-3β was decreased, whereas the β-catenin expression was increased in human endothelial cells treated with ellipticine. We concluded that ellipticine at sublethal dosage could be successfully repositioned as a pro-angiogenic substance by in silico virtual screening.

## 1. Introduction

Angiogenesis, the formation of new blood vessels, is critically involved in many normal physiological processes, such as embryonic development, ovulation, and wound healing, and is a critical step in many pathological conditions [1]. In many patients with coronary arterial diseases, peripheral arterial diseases (such as arteriosclerosis obliterans), thromboangiitis obliterans, and collagen disease, the anatomic extent and distribution of arterial occlusive disease make the patients unsuitable for surgical or percutaneous revascularization. Additionally, the diseases sometimes follow an inexorable downhill course. However, no pharmacological treatment has shown a favorable effect in the natural history of critical limb ischemia in those patients [2]. Collectively, it is urgent to develop an effective strategy to salvage ischemic tissues using a novel compound.

Drug repurposing (also known as drug repositioning, reprofiling, or retasking) involves finding novel medical uses for existing drugs, including approved, investigational, discontinued, and shelved therapeutics [3]. In drug discovery, drug repurposing has several advantages compared with de novo drug discovery because the new therapeutic indications, including cost-effectiveness and time-saving with less risk, are built on already available pharmacokinetics, as well as toxicological and manufacturing data, leading to therapeutic solutions in an expedited manner [4]. Furthermore, it helps in circumventing preclinical development and optimization issues, making drug discovery more efficient and less expensive. Alternatively, advances in drug discovery have dramatically increased the number of synthetic and naturally sourced molecules available for testing in vitro in biochemical and cellular assays [5]. In addition, recent advancements in virtual screening methods extend the possibilities to molecules that do not necessarily exist physically in an investigator’s collection but can be readily obtained through purchase or synthesis. The other advantage is that as a result of the computational prediction of binding affinity, only a relatively small subset of compounds requires testing, and therefore, activities can be quantified in a low- or medium-throughput assay format [6]. A method that can be achieved by combining drug repurposing and computational manners in drug discovery is called “in silico repurposing.”

Ellipticine is an alkaloid extracted from the trees of the species *Ochrosia elliptica**,* which inhibits the enzyme topoisomerase II via intercalative binding to DNA replication [7]. In clinical trials, ellipticine derivatives have been observed to induce remission of tumor growth but are not used for medical purposes because of their high toxicity. Side effects include nausea and vomiting, hypertension, cramps, pronounced fatigue, mouth dryness, and mycosis of the tongue and esophagus [8]. Although there are many studies of the antiproliferative properties of several topoisomerase II inhibitors (etoposide and amsacrine), including ellipticine, no studies have focused on the influence of this molecule in regulating angiogenesis.

This study shows that ellipticine—a drug identified by the virtual screening strategy using the LINCS L1000CDS^2^ perturbation dataset, chemical similarity, and molecular docking test—enhances angiogenesis in in vitro and ex vivo models at a noncytotoxic dosage, and was not used as an anticancer drug. Thus, the results of this study provide valuable insight into a drug development process that can regenerate the ischemic tissue by provocating angiogenesis.

## 2. Results

### 2.1. Initial Identification of Candidate Drugs to Enhance Angiogenesis by In Silico Virtual Screening

To screen the LINCS L1000CDS^2^ perturbation database to identify candidate drugs for provocating angiogenesis, we collected five angiogenesis microarray studies from the Gene Expression Omnibus (GEO) data repository (Table 1). Each dataset was analyzed with GEO2R to obtain gene expression change data during angiogenesis. All differentially expressed gene symbols from five microarray results were merged into 749 unique differentially expressed gene (DEG) symbols and used for virtual screening. Next, we searched the L1000CDS^2^ to prioritize small molecules predicted to be either reverse or mimic expressions of the DEG signatures [9]. The L1000CDS^2^ calculated the pairwise cosine distance between the directions of the disease-drug characteristics and provided ranked lists of scores for the candidate compounds. We obtained 347 perturbations of each (Supplemental Appendix A). By thoroughly reviewing each compound, we identified that a glycogen synthase kinase-3β (GSK-3β) inhibitor, TWS-119 (4,6-disubstituted pyrrolo-pyrimidine), was well known to have the pro-angiogenic effect in previous studies [10,11,12]. Hence, we focused on TWS-119 for further screening.

We used hierarchical and multidimensional clustering analyses to identify the chemicals that have a structural similarity to TWS-119 among the first-screened compound libraries, as implemented in the ChemMine software suite [13]. Hierarchical clustering showed that 18 compounds had a similar structure within the respective libraries. Among the 18 compounds, we found that five compounds were overlapped in multidimensional clustering analysis [14]. The result of the structural similarity test is shown in Figure 1. The general description of the five chemicals is shown in Table 2.

Finally, we performed molecular docking using AutoDock v.4 on GSK-3β (PDB: 3I4B) for five compounds (TWS-119, ellipticine, BRD-K39687892, SB-334867, and CHEMBL2142877) because TWS-119 was known as a GSK-3β inhibitor. These tested inhibitors showed negative binding energy. Ellipticine (except for CHEMBL2142877, which cannot be purchased) showed the highest binding affinity (−7.8 kcal/mol) than did TWS-199 (−7.5 kcal/mol) and other compounds (BRD-K39687892, −7.1 kcal/mol; SB-334867, −6.9 kcal/mol; and ATP, −4.6 kcal/mol) (Figure 2).

### 2.2. The Noncytotoxic Concentration of Ellipticine in a Dose–Response Curve Is 156.25 nM

The efficacy of ellipticine was verified in vitro to confirm that virtual screening was useful. It was tested in human umbilical vein endothelial cells (HUVECs) for 24 h using the cell counting kit-8 (CCK-8) assay. Analysis of the cell viability assay by individual dose ellipticine using twofold serial dilutions (from 10,000 to 78.125 nM) is presented in Figure 3. Because ellipticine, a topoisomerase II inhibitor, is an anticancer drug, a cell viability assay was used to measure appropriate drug concentrations that do not cause cell death. Thus, a concentration of 156.25 nM was used for the following cellular experiments.

### 2.3. Low-Dose Ellipticine Reduces the Phosphorylation of GSK-3β and Increases β-Catenin Expression

We examined the activity of the candidate compound ellipticine using western blotting and immunofluorescence assays. Western blotting (Figure 4A) revealed that 156.25 nM ellipticine decreased the phosphorylation of GSK-3β. By contrast, it enhanced β-catenin protein expression after treatment for 8 h. Immunofluorescence assay (Figure 4B) was used to detect the status of β-catenin in HUVECs. Following a challenge with 156.25 nM ellipticine for 8 h, the signals of β-catenin expression in the ellipticine-treated group were significantly enhanced compared to the control group.

### 2.4. Low-Dose Ellipticine Enhances Endothelial Cell Migration

We performed an HUVEC angiogenesis assay (Figure 5A,B). HUVECs treated with ellipticine showed a significantly greater number of tubes and branch points than the control (DMSO treatment). Additionally, we tested the effect of ellipticine on HUVEC collective bidimension through the wound-healing assay. We found a significantly greater increment in wound closure after 8 h of incubation in ellipticine-treated cells with respect to control cells (Figure 5C,D). Furthermore, cell migration was enhanced with the presence of low-dose ellipticine.

### 2.5. Low-Dose Ellipticine Has a Pro-angiogenic Property

We analyzed the capillary plexus of the treated embryos to determine whether low-dose ellipticine possesses pro-angiogenic properties. In the control embryos, a normal vasculature was seen around the embryo. In ellipticine-treated embryos, changes were seen in the vascular pattern (Figure 6A). The number of branches, total tube length, and total branching length were presented in percentage. The number of branch area and total branching length were significantly higher in ellipticine-treated embryos than in the control (*p* < 0.05 and *p* < 0.001, respectively) (Figure 6B). With regard to in vivo vessel formation, chorioallantoic membrane (CAM) assay with low-dose ellipticine resulted in significantly greater vessel density, total vessel network length, and total branch points than it without ellipticine.

## 3. Discussion

Efforts to stimulate tissue vascularization in patients with ischemia resulting from a coronary or peripheral arterial disease [15,16] have passed through several scientific phases over the last dozen years based on our increased understanding of the molecular mechanisms underlying vascular homeostasis [17]. This study shows that ellipticine, a drug identified by a virtual screening strategy using the LINCS L1000CDS^2^ perturbation dataset, ChemMine, and AutoDock, increased angiogenesis in in vitro and ex vivo models at low concentrations. Furthermore, it provides evidence that depending on the concentration, ellipticine regulates angiogenesis and confirms the pro-angiogenic ability of low-dose ellipticine. Thus, a low concentration of ellipticine, such as around 156.25 nM, enhanced capillary-like formation by increasing cell proliferation and migration, which can induce neovascularization by inhibiting the phosphorylation of GSK-3β and upregulation of β-catenin expression. Interestingly, although ellipticine, a topoisomerase II inhibitor, could be used as an antitumor molecule, its low dose in the nanomolar range (156.25 nM) could offer crucial therapeutic perspectives for treating and preventing ischemic diseases. Furthermore, treatment with low-concentration ellipticine might have potentially no serious adverse effects and a high degree of tolerability with a good safety profile.

Drug repurposing is an efficient approach to finding new indications by exploiting available and approved drugs. We do not need to test them all over again because there are existing data on pharmacokinetics and pharmacodynamics, and we could skip preclinical studies and ultimately phase 1 of the clinical phase. Additionally, computational methods, such as structure-based drug design, can make drug discovery more resourceful and accurate because it allows for screening large compound libraries for specific targets quickly. Although finding new protein targets from the previously approved drugs is challenging, computational techniques, such as virtual screening for drug repurposing, have become one of the most widely used methods to make the drug development process more efficient, faster, and less expensive. The use of the virtual screening of the connectivity map methodology based on the integration of the available gene expression data helps in identifying new indications for recently certified drugs [18,19,20]. Using this approach, therapeutic candidates for various diseases, such as dyslipidemia [21], pain [22], influenza [23], lung cancer [24], and renal cancer [25], have been identified. Moreover, the methodological validity of the connectivity map has been confirmed as a promising technique for drug development and repositioning [19].

The double-stranded nature of DNA creates a special set of problems for processing that requires strand unwinding, such as transcription and replication. The unwinding during these processes creates a topological problem because unwinding must be compensated by overwinding elsewhere in the DNA molecule. Thus, there is a clear requirement to alter DNA topology by introducing transient double-strand breaks; only DNA topoisomerase II can perform this reaction and is essential for all eukaryotic cells [26]. Ellipticine, an alkaloid isolated from *Apocynaceae* plants, exhibits significant antitumor activities, which inhibittopoisomerase II via intercalative binding into DNA [7]. Many studies have shown that this antitumor agent forms covalent DNA adducts after enzymatic activation with peroxidases and cytochrome P450 [27,28], suggesting additional DNA-damaging effects of ellipticine. Furthermore, it has functioned through multiple mechanisms that participate in cell cycle arrest by regulating the expression of cyclin B1 and Cdc2 and phosphorylation of Cdc2 [29] to induce apoptotic cell death by generating cytotoxic-free radicals, activating the Fas/Fas ligand system, and regulating Bcl-2 family proteins [29,30,31]. Moreover, ellipticine is reported to trigger the apoptosis of human endometrial cancer cells [31,32], neuroblastoma cells [33], and rheumatoid arthritis fibroblast-like synoviocytes [34].

The IC_50_ value of ellipticine in various cancer cell lines was approximately 1 µM. Neuroblastoma IMR-32 cells, followed by neuroblastoma UKF-NB-4 and UKF-NB-3 and leukemia HL-60 cells, were the most sensitive to ellipticine, with IC_50_ values lower than 1 μM. When the sensitivity of other cells to ellipticine was compared, cytotoxicity of this agent to human breast adenocarcinoma MCF-7 cells and a glioblastoma U87MG cell line was comparable (the IC_50_ values were approximately 1 μM). By contrast, leukemia CCRF-CEM cells were less sensitive. However, the IC_50_ value for ellipticine was almost four times higher in these leukemia cells than in MCF-7 and U87MG cells [35].

According to ISO 10993-5, percentages of cell viability above 80% are considered noncytotoxic; within 80–60%, weak; 60–40%, moderate, and below 40%, strong cytotoxicity [36]. Thus, we have determined the concentration of ellipticine as 156.25 nM and conducted further experiments to verify the pro-angiogenic effect.

β-catenin is a central signaling molecule in the canonical Wnt pathway, an important regulatory system that controls cell fate, and when disturbed, it is a major driver of cancer. β-catenin is a genuinely multifunctional protein with two major cellular pools. First, it localizes at the plasma membrane as part of multiprotein cell–cell junction complexes (adherens junctions), and second, it is free in the cytosol or nucleus. The free form is a vital transcriptional regulator of specific target genes. However, in cells not stimulated with Wnt ligands, it is maintained at a low level by GSK-3β phosphorylation, which targets β-catenin for ubiquitination and degradation [37]. GSK-3β, a multifunctional Ser/Thr kinase, is a vital component of diverse signaling pathways regulating protein synthesis, glycogen metabolism, cell mobility, proliferation, and survival [38,39]. There are two mammalian GSK-3 isoforms encoded by distinct genes: GSK-3α and GSK-3β, which share 85% identity [40]. Despite a high degree of similarity and functional overlap, these isoforms are not functionally identical and redundant. It is known whether GSK-3β plays a central role in various signaling pathways, such as the Wnt/β-catenin, Hedgehog, Notch, and insulin signaling pathways [41,42,43,44,45]

The possible pro-angiogenic mechanism for low concentrations of ellipticine is as follows: β-catenin has a positive effect on the *serine/threonine-specific protein kinase-mechanistic target of rapamycin* (AKT-mTOR) pathway [37], thereby inducing endothelial nitric oxide synthase (eNOS) expression and increasing nitric oxide (NO) [46]. Additionally, an activated AKT enhances the expression of hypoxia-inducible factor-1α (HIF-1α), which has a great influence on the increase in vascular endothelial growth factor (VEGF) [47]. Eventually, both NO and VEGF play an essential role in promoting vascular regeneration. Conversely, because GSK-3β negatively regulates β-catenin and HIF-1α [41,42], low-dose ellipticine is thought to facilitate angiogenesis by inhibiting GSK-3β (Figure 7).

There are several limitations to this study. First, it is necessary to determine the appropriate dosage of drugs that enhances angiogenesis. Second, further studies on the specific mechanism of ellipticine pro-angiogenetic effect are needed. Nevertheless, our research is significant as the first drug repurposing results in the neovascular regeneration of the topoisomerase II inhibitor ellipticine using an in silico method.

In summary, the main finding of this study is the demonstration of the paradoxical properties of the potent topoisomerase II inhibitor (antitumor effect) ellipticine in regulating angiogenesis at a low dose by in silico virtual screening. Furthermore, this study indicated that a low concentration of ellipticine might improve angiogenic remodeling after ischemia. Thus, further experiments are needed to validate our findings.

## 4. Materials and Methods

### 4.1. Screening Procedure

The general chemical screening procedure is described in Figure 8. Gene Expression Omnibus (GEO) microarray datasets related to angiogenesis studies were collected for the first screening. The GEO2EnrichR web application and R packages were used to extract the characteristic direction (CD) to identify differentially expressed gene signatures. The CD method brings multivariate gene information to a single direction in hyperspace, which makes it possible to compare similar gene signature information. All microarray datasets were extracted with a cutoff of 500, and other options were default conditions. The extracted CD signatures from GEO2EnrichR were used as input data on L1000CDS^2^ to screen pro-angiogenic molecules. L1000CDS^2^ is a web-based search engine that includes CD signatures of thousands of small molecules.

ChemMine Tools and ChemminR, structural similarity search tools for small-molecule analysis and clustering, were used for the second screening. PubChem compound identifications (CIDs) of first-screened small molecules were imported to construct structural fingerprints. Hierarchical and multidimensional clustering was processed for the downstream analysis. The hierarchical clustering in ChemMine Tools organizes similar objects in a tree with branch lengths proportional to the compound-to-compound similarities defined using the Atom Pair descriptors [48] and the Tanimoto coefficiency [49]. The Tanimoto coefficient of similarity for Molecules A and B was defined as c/(a + b − c), where a is the number of on-bits in molecule A, b is the number of on-bits in molecule B, and c is the number of bits that are on in both molecules A and B. The Tanimoto coefficient ranges from 0 to 1, with higher values indicating greater similarity. The single linkage method was performed to define the distance between two clusters. The distance matrix, which is converted from similarity matrix by subtracting the Tanimoto similarity values from 1, was visualized on a heat map. Multidimensional clustering is an algorithm for embedding compounds in a Euclidian space such that the distances between compounds can preserve the dissimilarities between compounds. ChemMine Tools follows the analysis of Mardia [50], which is known as classical metric multidimensional scaling or principal coordinate analysis. Two dimensions were set for multidimensional clustering. Other options were processed in default conditions.

Next, to evaluate the physicochemical effect of the second-screened chemical, molecular docking simulation was conducted with AutoDock v.4. The structure data format files of chemicals were collected from PubChem and converted to a PDBQT format using Open Babel (v.2.3.1) and Raccoon (v.1.5.7rc1). GSK-3β crystal structure (PDB: 3I4B) was imported to AutoDock v.4 for the target protein. The water molecules that could have been included in the target protein.pdb files were removed, polar hydrogens were added to the proteins, and Gasteiger charges were added as previously described [51]. The target protein was changed to a PDBQT, and each chemical PDBQT was imported from AutoDock v.4. The grid box (size: 40Å × 40Å × 40Å) was set on the binding site of the target protein. Other options were set with default conditions for the docking process. The rank of the binding energy (−kcal/mol) was used as the main criteria to select final molecules for in vitro analysis.

### 4.2. Chemicals

On the basis of the result of the screening procedure, ellipticine was purchased from TargetMol (Target Molecule Corp., Boston, MA, USA). Unfortunately, CHEMBL214877 could not be purchased.

### 4.3. Cell Culture and Chemical Treatment

HUVECs were purchased from Lonza. Cells were incubated with EGM-2 medium (Lonza) at 37 °C with 5% CO_2_ in a humidified incubator. A passage between two and six was used for the experiments. Each chemical was dissolved in DMSO. The selected chemical concentration was mixed with EGM-2 medium and treated on cells. The medium was replaced every 2–3 d. The concentration of DMSO in the medium was not more than 0.2% in the experiments.

### 4.4. Cell Viability Assay

To analyze the cytotoxic effect of each chemical on HUVECs, CCK-8 (Dojindo, Rockville, MD, USA) was used according to the manufacturer’s instructions. HUVECs (5 × 10^3^ cells) were plated in noncoated 96-well plates and cultured for 24 h. Various doses of the chemical or DMSO (control) were treated on each well. Following 24 h of incubation, 10 μL of CCK-8 solution was added to each well for 2 h. Absorbance at 450 nm was detected using a microplate reader (Synergy H1, Biotek, Vermont, USA) to determine cytotoxicity.

### 4.5. Western Blot

HUVECs were lysed using PRO-PREP protein extraction solution (iNtRON Biotechnology, Seongnam, Korea). Equal amounts of proteins (15–30 μg) were loaded on 10% gel and separated by SDS-PAGE. The separated proteins were transferred to a nitrocellulose membrane. Following 1 h of blocking with 5% nonfat dry milk in TBS-T (0.2% TWEEN-20 with TBS), 1:1000 PBS-T diluted primary antibody was treated and incubated overnight at 4 °C. After 10 min and three times of TBS-T washing, secondary antibodies diluted at 1:2000 in TBS-T were incubated for 2 h. Antibodies against phosphorylated GSK-3β (Ser9) (cat. no. 9336), β-Catenin (D10A8) (cat. no. 8480), and β-actin (cat. no.4970) were purchased from Cell Signaling Technology, Inc (Danvers, MA, USA). Horseradish peroxidase (HRP)-conjugated anti-rabbit IgG (cat. no. A120-101P) secondary antibodies were purchased from Bethyl Laboratories, Inc (Montgomery, TX, USA). Immunoreactive proteins were visualized with an ECL detection kit (Pierce ECL, Thermo Fisher Scientific, Waltham, MA, USA).

### 4.6. Immunofluorescence

HUVECs were plated and cultured on slides. Following chemical-treated incubation for 8 h, cells were fixed with 4% paraformaldehyde for 10 min. After washing three times with PBS, 0.1% Triton X-100 in PBS was loaded on cells for 10 min and washed three times with PBS. BSA (1%) in PBS-T (0.1% TWEEN-20 with PBS) was used as a blocking solution for 30 min. The slides were incubated with PBS-T-diluted to 1:100 primary β-Catenin (D10A8) (cat. no. 8480) antibody overnight at 4 °C, followed by three times and 5 min of washing with PBS-T. Incubation with secondary diluted to 1:400 Alexa Fluor 488 goat anti-rabbit (cat. no. ab150077) antibodies was performed for 1 h at RT in a dark environment. After three 5-min washes, 1 μg/mL of DAPI (4′,6-diamidino-2-phenylindole) (cat. no. ab228549) was used for counterstaining for 1 min, followed by rinsing with PBS. The mounted slide was visualized using a confocal microscope (Zeiss LSM880).

### 4.7. Tube Formation Assay

A tube formation assay was performed to evaluate the effect of chemicals on angiogenic properties in HUVEC. Matrigel (60 μL) thawed at 4 °C was coated on each well of 96-well plates, followed by 30 min incubation at room temperature (20~22 °C) for initial solidification. Then, 2 h of additional solidification was performed in a humidified incubator. HUVECs (1.5 × 10^4^/well) were seeded in Matrigel-coated 96-well plates. After 2–3 h of incubation at 37 °C for initial tube formation, predetermined concentrations of chemicals from the CCK assay with EGM-2 medium were treated on HUVEC for 8 h. An assessment of tube network was performed under a fluorescence microscope. Calcein AM (1 mM; BD Biosciences, Franklin Lakes, NJ, USA) was treated 20 min before the assessment. The number of branching points was calculated using ImageJ (NIH).

### 4.8. Migration Assay

To analyze the migration effect of chemicals on HUVECs, a migration assay was performed using a wound-healing assay multiwell plate (ibidi, Fitchbug, WI, USA) according to the manufacturer’s instructions. Briefly, 70 μL of 1.0 × 10^5^ HUVEC was plated on each well’s insert. After observing confluency within 1 to 2 d of incubation, predetermined concentrations of chemicals or DMSO with EGM-2 medium were treated on each well, followed by insert removal with sterile tweezers. The assessment was performed after 8 h of incubation under a fluorescence microscope. Calcein AM was treated 20 min before the assessment. The percentage of wound closure was determined by ImageJ.

### 4.9. Chorioallantoic Membrane (CAM) Assay

Fertilized chicken eggs (Ross 308) were incubated at 37 °C in 60–80% humidity for 24 h. Eggshells were cleaned with 70% ethanol, and a pinpoint hole was then made through the blunt pole of the shell. Next, a sterilized Thermanox coverslip (Nunc, Thermo Fisher Scientific, Waltham, MA, USA) saturated with either sterile phosphate-buffered saline or ellipticine was placed on the CAM. A predetermined concentration of chemicals was applied for 72 h. The holes were then sealed with cellophane tape, and the eggs were resubjected to incubation under the abovementioned conditions. On Hamburger–Hamilton developmental stage 22–24 (day 4 of the incubation period), the eggshells and shell membranes were aseptically removed to expose the surface of the CAM by peeling a 25 × 25 mm window on the egg. High-quality images (4000 × 3000 pixels) were captured using a stereomicroscope (Luxeo 4D Stereozoom Microscope, Labomed, CA, USA) and analyzed. Blood vessel density was quantified using ImageJ and represented as a bar diagram.

### 4.10. Statistical Analyses

Statistical analysis of quantitative data was performed using R (v.3.6.3; The R Foundation for Statistical Computing, Vienna, Austria; http://www.R-project.org/, accessed 15 April 2020). A two-tailed Student’s *t*-test was used for single-group comparison. *p* < 0.05 between experimental groups was considered statistically significant.

## Figures and Tables

**Figure 1 ijms-22-09067-f001:**
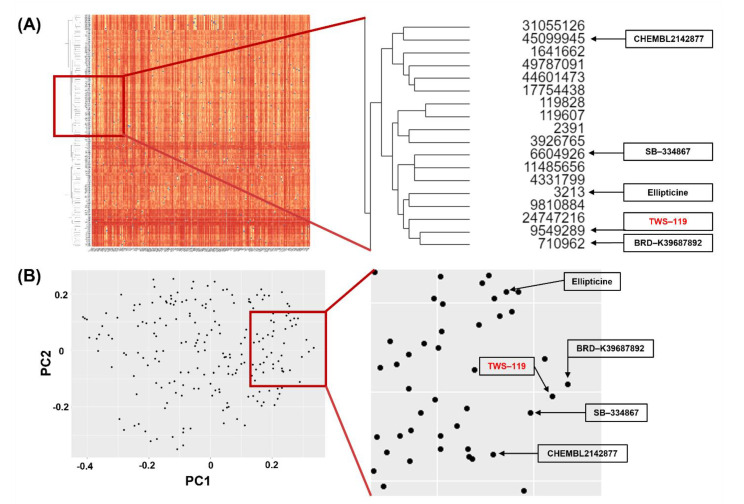
Results of structural similarity; (**A**) hierarchical clustering and (**B**) multidimensional clustering. The hierarchical clustering method repeats the process of finding the two most similar structures, then aggregating into the same cluster until all rows are clustered. After the repetition, structurally similar chemicals are arranged together to generate dendrogram. A multidimensional clustering technique is used to visualize high dimensional data into two-dimensional space. This method transforms the coordinate vectors to explain the highest two variances of the dataset. Hierarchical clustering showed that 18 compounds had a similar structure within the respective libraries. Among the 18 compounds, we found that five compounds were overlapped in multidimensional clustering analysis.

**Figure 2 ijms-22-09067-f002:**
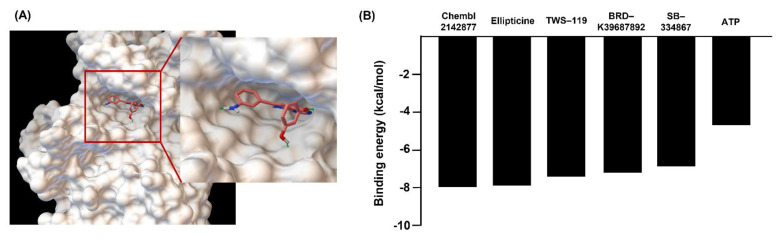
Results of GSK-3β chemical docking test. Small molecules that putatively bind the binding box of GSK-3β were identified using an in silico docking approach. We identified five small molecules that putatively bind the binding box of GSK-3β. We conducted the binding energy calculation for the five compounds using AutoDock v.4 (**A**). Results were ranked according to the calculated binding affinity (**B**). The lowest binding affinity docking pose for ellipticine/GSK-3β was superimposed onto the ATP/GSK-3β crystal structure.

**Figure 3 ijms-22-09067-f003:**
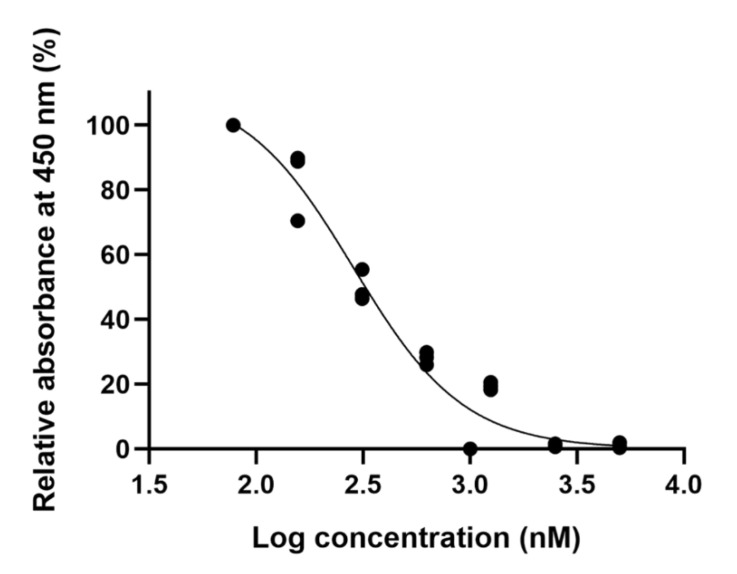
Analysis of cell viability in human umbilical vein endothelial cells (HUVECs). Cell viability was analyzed using Cell Counting Kit-8. The HUVECs were treated with 78.125–10,000 nM ellipticine for 24 h. The noncytotoxic concentration of ellipticine (<156.25 nM) was obtained. Dose–response curves for ellipticine at fixed substrate concentrations. Dotted lines represent the best fit of experimental data to the four-parameter Hill equation.

**Figure 4 ijms-22-09067-f004:**
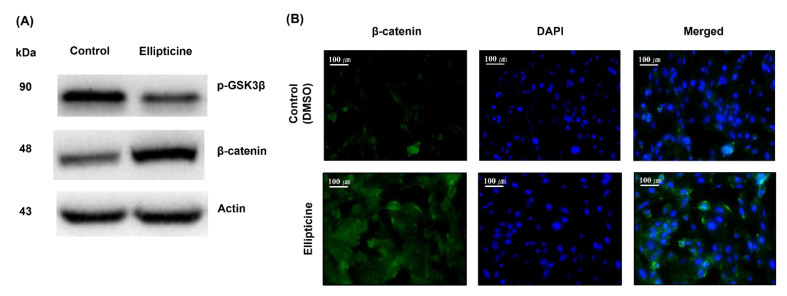
Evaluation of the activity of the candidate chemical ellipticine. HUVECs were treated with 156.25 nM ellipticine for 8 h. After the treatments, the cell lysates were extracted and the levels of phosphorylated GSK-3β and β-catenin expression were analyzed by western blotting using specific antibodies (**A**). Actin control was included. Immunofluorescence assay revealed that the expression of β-catenin (green) in ellipticine-treated cells was significantly enhanced compared to the control. Nuclei were counterstained with DAPI (blue) (**B**). (40× magnification, scale bar = 10 μm).

**Figure 5 ijms-22-09067-f005:**
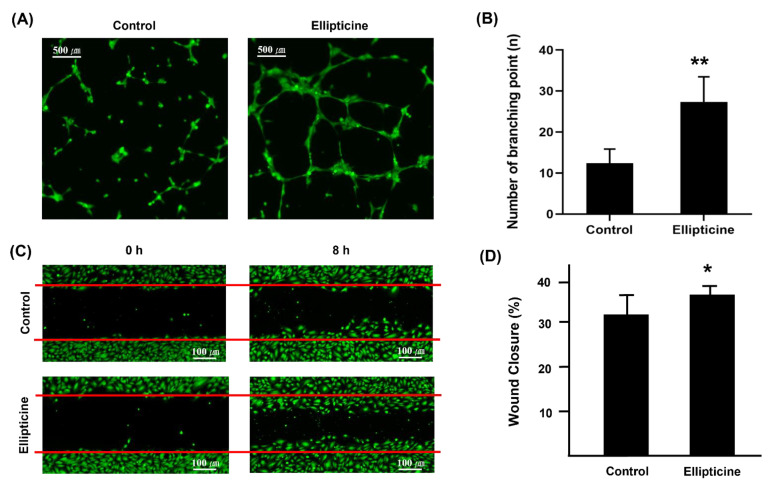
In vitro angiogenesis assay using HUVECs. HUVECs tube-forming assay (**A**) measured by the number of junctions. DMSO was used as a negative control (40× magnification, scale bar = 500 μm). It reveals reassembly of endothelial cells (green dot) and formation of new cell-cell contacts, vessel lumina, and cell network formation (green line) in ellipticine-treated cells were significantly enhanced as compared with control. Graph (**B**) shows significant activation of endothelial tube formation by ellipticine at 156.25 nM. Confluently cultured endothelial cells were wounded using a sterile tip and treated without or with 156.25 nM ellipticine for 8 h (**C**) (40× magnification, scale bar = 100 μm). There is a significant difference in the amount and velocity of vascular endothelial cells (green dot) migrating from both sides to fill the gap in the wound (red line). The wound closure levels are shown (**D**). * *p* < 0.05, ** *p* < 0.01.

**Figure 6 ijms-22-09067-f006:**
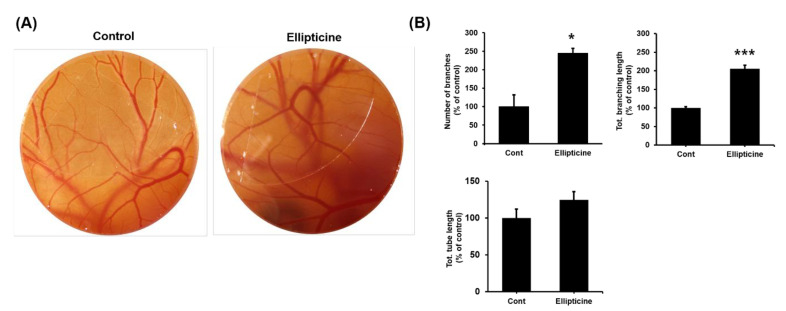
Chorioallantoic membrane (CAM) assay for pro-angiogenic effect. CAMs were treated with either 156.25 nM ellipticine or DMSO as a control for 96 h (**A**). The number of branches, total branch length, and total tube length were measured and shown (**B**). * *p* < 0.05, *** *p* < 0.001.

**Figure 7 ijms-22-09067-f007:**
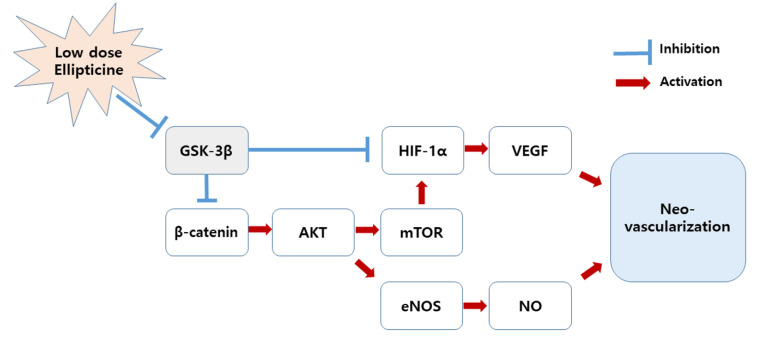
A possible pro-angiogenic mechanism for a low concentration of ellipticine. Low-dose ellipticine inhibits GSK-3β, thus upregulating β-catenin and HIF-1α. This ultimately promotes blood vessel regeneration. AKT, serine/threonine-specific protein kinase; *eNOS*, endothelial nitric oxide synthase; GSK-3β, glycogen synthase kinase-3β; HIF-1α, hypoxia-inducible factor-1α; mTOR, *mechanistic target of rapamycin*; NO, nitric oxide; VEGF, vascular endothelial growth factor.

**Figure 8 ijms-22-09067-f008:**
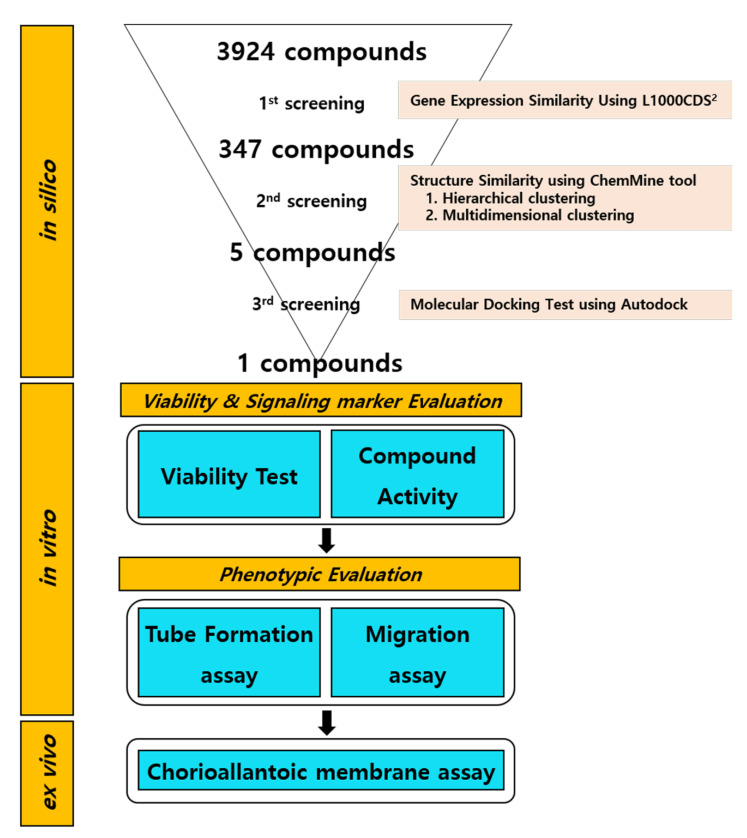
Summary of the study design.

**Table 1 ijms-22-09067-t001:** List of GEO datasets used in this study.

GEO Accession Number	Experimental Group	Sample Number	Species	Study Summary
GSE837	High serum PlGF 24 h High serum VEGF-A 24 h Low serum PlGF 24 h Low serum VEGF-A 4 h Low serum VEGF-A 24 h	GSM12894–GSM12926	*Homo sapience*	HUVECs are treated with the angiogenic factors VEGF-A and PlGF in low or high serum media
GSE9677	Angiopoietin-1 (Sparse)Angiopoietin-1 (Confluence)	GSM244647–GSM244654	*Homo sapience*	Angiopoietin-1 stimulation on HUVECs
GSE19098	VEGF (spread) VEGF (unspread)	GSM473026–GSM473037	*Homo sapience*	50 ng/mL VEGF or no growth factor was treated on HUVEC for 18 h
GSE22695	VEGF 30 min VEGF 60 min VEGF 150 min	GSM385333–GSM385353	*Homo sapience*	HUVEC and CEP were treated with VEGF-A for 30, 60, 150 min
GSE71216	VEGF	GSM1830134–GSM1830137	*Homo sapience*	HUVECs were treated with 16 ng/mL VEGF165 for 4 days

**Table 2 ijms-22-09067-t002:** List of chemicals of interest.

Compounds	IUPAC Name	Known MOA	CID	MW
TWS-119	3-[[6-(3-aminophenyl)-7H-pyrrolo[2,3-d]pyrimidin-4-yl]oxy]phenol	GSK-3β inhibitor	9549289	318.3
Ellipticine	5,11-dimethyl-6H-pyrido[4,3-b]carbazole	Topoisomerase II inhibitor	3213	246.31
BRD-K39687892	N-(3-fluorophenyl)-2-pyridin-4-ylquinazolin-4-amine	GSK-3β inhibitor	710962	316.3
SB-334867	1-(2-methyl-1,3-benzoxazol-6-yl)-3-(1,5-naphthyridin-4-yl)urea	Orexin antagonist	6604926	319.32
CHEMBL2142877	4-phenyl-N-(6-phenyl-3-pyridin-2-yl-1,2,4-triazin-5-yl)-1,3-thiazol-2-amine	Unknown	45099945	408.5

## Data Availability

All data reported in the manuscript.

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
