# Peer review of "Paradoxical Pro-angiogenic Effect of Low-Dose Ellipticine Identified by In Silico Drug Repurposing"

_ijms, 2021, doi:10.3390/ijms22169067_

Round 1
Reviewer 1 Report
Inadequate vessel maintenance or growth causes ischemic diseases and developing an effective strategy to salvage ischemic tissues using a novel compound is urgent. The authors found a novel drug candidate for pro-angiogenesis by in silico virtual screening in the presented work. This is an interesting work and should be suitable for the International Journal of Molecular Sciences. The reviewer has some questions.
1. Figure 1. The authors described they used 3924 compounds at the beginning of in silico screening, and only one compound was left after the screening. But there are still in vitro and ex vivo tests. Only one compound for these two tests, how to make sure it would work?
2. Figure 2. Chemical docking test. The authors only used five compounds for the docking test. And after the docking test, only one compound was left. This result doesn't convince everyone. This bath is too less.
3. Figure 5 and 6. It is better to provide a scale for images.
Author Response
We greatly appreciate your detailed and constructive comments. Below we provide a point-to-point response to your comments and clarify the important points of your main concerns.
â–£ Reviewer #1
- Figure 1. The authors described they used 3924 compounds at the beginning of in silico screening, and only one compound was left after the screening. But there are still in vitro and ex vivo tests. Only one compound for these two tests, how to make sure it would work?
==> The 3924 compounds were simulated as scientifically and objectively as possible in three stages. And GEO data is an open database that has been experimented one by one and released the results. Our researchers studied these previous findings in depth and discussed them a number of times. And then, we chose one substance that was considered the most likely based on the docking test results. And we demonstrated through in vivo, ex vivo experiments that ellipticine, known as an anti-cancer drug, has a proangiogenic effect in low concentration.
- Figure 2. Chemical docking test. The authors only used five compounds for the docking test. And after the docking test, only one compound was left. This result doesn't convince everyone. This bath is too less.
==> We selected five substances closest to TWS119 based on chemical structure similarity before conducting the docking test. There was not only one left after the docking test, but all five were less than ATP binding energy, all of which were candidate substances. However, since chembl2142877 was not available on the market, we conducted an experiment with ellipticine, which has the second lowest binding energy. In the end, our choices came with good results, and we were able to achieve a successful drug repurposing.
- Figure 5 and 6. It is better to provide a scale for images
==> We put a scale that fits each figure.
Reviewer 2 Report
Drug repositioning is the fastest, most sustainable, and least expensive process for identifying drug candidates for the treatment and prevention of a disease of interest because it relies on compounds that have already undergone pre-clinical safety studies.
The authors in the manuscript present the identification of a pro-angiogenic drug that could offer therapeutic perspectives for ischemic diseases, by repositioning the intercalative binder of DNA, inhibiting DNA replication and transcription, ellipticine at a sublethal dose.
The virtual screening in silico of large libraries of commercial compounds to be repositioned is the initial step for the identification of molecules that can bind a clinical target of interest in a pocket that could be critical for its activity. To identify pro-angiogenic drug candidates, the authors used a very interesting signature-based virtual screening strategy that includes the incorporation of available high-throughput gene expression data. First, the authors identified DEGs during angiogenesis. Then they identified small molecules predicted to be either reverse or mimic expressions of the DEG signatures, so focusing on the pro-angiogenic GSK3 inhibitor TWS-119. Then the authors used hierarchical and multidimensional clustering analyses, to identify the chemicals that have structural similarity with TWS-119, and focused on five chemicals, including ellipticine. Docking results allowed them to focus on this last compound for the experimental analyses.
In my opinion, the least clear step of the manuscript is the one related to hierarchical and multidimensional clustering analysis, whose methods should be better explained for non-experts, like me, who have an experimental background.
The experimental part is very clear, even if in the materials and methods section some details concerning the reagents used are missing, such as the primary and secondary antibodies used in western blotting and immunofluorescence analyses. In the materials and methods “PADI” has to be changed with “DAPI” and some detail concerning the stained structures should have to be added in the legend of Figures 5 and 6 in order to make clear fluorescence results also for the non-experts.
Overall, the manuscript is well written, and the rationale of the study is clear. The drug repurposing approach is very interesting. The results are clearly described, and Introduction and Discussion are well supported by literature data. In the discussion also the criticisms of the study are reported as the need to further investigate the specific mechanism of ellipticine pro-angiogenetic effect.
In my opinion, the work with some little informative integration is suitable for publication.
Best regards
Author Response
We greatly appreciate your detailed and constructive comments. Below we provide a point-to-point response to your comments and clarify the important points of your main concerns.
â–£ Reviewer #2
- In my opinion, the least clear step of the manuscript is the one related to hierarchical and multidimensional clustering analysis, whose methods should be better explained for non-experts, like me, who have an experimental background.
==> Thank you for your advice. According to your opinion, we put the following on page 4 in the manuscript and the detailed in the legend of Figure 2.
==> The hierarchical clustering in ChemMine Tools organizes similar objects in a tree with branch lengths proportional to the compound-to-compound similarities defined using the Atom Pair descriptors [9] and Tanimoto coefficiency [10]. Tanimoto coefficient of similarity for Molecules A and B was defined as c/(a+b-c), where a is the number of on bits in molecule A, b is the number of on bits in molecule B and c is the number of bits that are on in both molecules A and B. Tanimoto coefficient ranges from 0 to 1 with higher values indicating greater similarity. Single linkage method was performed as how the distance between two clusters are defined. The distance matrix, which is converted from similarity matrix by subtracting the tanimoto similarity values from 1, visualized on a heat map. Multidimensional clustering is an algorithm for embedding compounds in a Euclidian space such that the distances between compounds can preserve the dissimilarities between compounds. ChemMine Tools follows the analysis of Mardia [11], which is known as classical metric multidimensional scaling or principal coordinate analysis. The two-dimensions was set for multidimensional clustering.
[9] R.E. Carhart, D.H. Smith, R. Venkataraghavan, Atom Pairs as Molecular-Features in Structure Activity Studies - Definition and Applications, J Chem Inf Comp Sci, 25 (1985) 64-73.
[10] X. Chen, C.H. Reynolds, Performance of similarity measures in 2D fragment-based similarity searching: Comparison of structural descriptors and similarity coefficients, J Chem Inf Comp Sci, 42 (2002) 1407-1414.
[11] K.V. Mardia, Some properties of clasical multi-dimesional scaling, Communications in Statistics - Theory and Methods, 7 (2009) 1233-1241
==> Figure 2. Results of structural similarity; (A) hierarchical clustering and (B) multidimensional clustering. Hierarchical clustering method repeats the process of finding two most similar structures then aggregating into the same cluster until all rows are clustered. After the repetition, structurally similar chemicals are arranged together to generate dendrogram. Multidimensional clustering technique is used to visualize high dimensional data into two-dimensional space. This method transforms the coordinate vectors to explain the highest two variance of dataset. Hierarchical clustering showed that 18 compounds had a similar structure within the respective libraries. Among the 18 compounds, we found that five compounds were overlapped in multidimensional clustering analysis.
- The experimental part is very clear, even if in the materials and methods section some details concerning the reagents used are missing, such as the primary and secondary antibodies used in western blotting and immunofluorescence analyses. In the materials and methods “PADI” has to be changed with “DAPI” and some detail concerning the stained structures should have to be added in the legend of Figures 5 and 6 in order to make clear fluorescence results also for the non-experts.
==> 1. Following your comment, we put reagents and antibodies in materials and methods section.
==> 2. We corrected it. PADI à DAPI (4′,6-diamidino-2-phenylindole)
==> 3. According to your comment, we revised the legends of figure 5 and 6.
Figure 5. Evaluation of the activity of the candidate chemical ellipticine. HUVECs were treated with 156.25 nM ellipticine for 8 h. After the treatments, the cell lysates were extracted and the levels of phosphorylated GSK-3β and β-catenin expression were analyzed by western blotting using specific antibodies (A). Actin control was included. Immunofluorescence assay revealed that the expression of β-catenin (green) in ellipticine-treated cells was significantly enhanced as compared with control. Nuclei were counterstained with DAPI (blue) (B) (40â…¹ magnification, scale bar = 100㎛).
Figure 6. In vitro angiogenesis assay using HUVECs. HUVECs tube-forming assay (A) measured by the number of junctions. DMSO was used as a negative control (40ⅹ magnification, scale bar = 500㎛). It reveals reassembly of endothelial cells (green dot) and formation of new cell-cell contacts, vessel lumina and cell network formation (green line) in ellipticine-treated cells were significantly enhanced as compared with control. Graph (B) shows significant activation of endothelial tube formation by ellipticine at 156.25 nM. Confluently cultured endothelial cells were wounded using a sterile tip and treated without or with 156.25 nM ellipticine for 8 h (C) (40ⅹmagnification, scale bar = 100㎛). There is a significant difference in the amount and velocity of vascular endothelial cells (green dot) migrating from both sides to fill the gap in the wound (red line). The wound closure levels are shown (D). *p < 0.05, **p < 0.01, ***p < 0.001.